# The outcomes of patients with kidney failure due to focal segmental glomerulosclerosis (FSGS) in Australia and New Zealand: A cohort study using the Australia and New Zealand Dialysis and Transplant Registry (ANZDATA)

Bhadran Bose [1,2,3]*, Elasma Milanzi[1], Elaine M. Pascoe[1,4], David W. Johnson[1,3,4,5], Sunil V. Badve[1,3,6]

1 Australasian Kidney Trials Network, The University of Queensland, Queensland, Australia, 2 Department of Nephrology, Nepean Hospital, Kingswood, Australia, 3 Australia and New Zealand Dialysis and Transplant (ANZDATA) Registry, Adelaide, Australia, 4 Centre for Kidney Disease Research, The University of Queensland, Queensland, Australia, 5 Division of Nephrology, Princess Alexandra Hospital, Brisbane, Australia, 6 Department of Nephrology, St George Hospital, Sydney, Australia

* bhadran.bose@health.nsw.gov.au

## Abstract

### Background

The outcomes of patients with focal segmental glomerulosclerosis (FSGS) on kidney replacement therapy (KRT) have not been well described. This study evaluated the outcomes of patients with kidney failure due to FSGS on KRT including dialysis and kidney transplantation.

### Method and materials

All adult patients with kidney failure who commenced KRT in Australia and New Zealand from 15[th] of May 1963 to 31[st] of December 2018 were retrospectively extracted from the Australia and New Zealand Dialysis and Transplant (ANZDATA) Registry. Outcomes of patients with FSGS were compared to those with other causes of kidney failure (non-FSGS).

### Results

85,052 patients commenced KRT during the study period, of whom 2991 (3.5%) were patients with FSGS. Compared to patients with non-FSGS, patients with FSGS experienced similar mortality on dialysis (adjusted hazard ratio [aHR] 0.98, 95% CI 0.90–1.06, p = 0.55) and following kidney transplantation (aHR 0.92, 95% CI 0.73–1.15, p = 0.47). The risk of first kidney allograft loss was higher in patients with FSGS (aHR 1.20, 95% CI 1.04–1.37, p = 0.01). However, when death was analysed as a competing risk, the survival in both groups was similar (sub-hazard ratio [SHR] 1.09, 95% CI 0.94–1.28, p = 0.26). Patients with FSGS had a longer waiting time for kidney transplantation (aHR 0.92, 95% CI 0.86–0.98,

**Data Availability Statement:** All relevant data are within the manuscript and its Supporting information files.

**Funding:** The authors received no specific funding for this work.

**Competing interests:** The authors have declared that no competing interests exist.

p = 0.02) and experienced an increased risk of disease recurrence in the allograft (aHR 1.73, 95% CI 1.35–2.21, p<0.001). Compared to patients with other forms of glomerular disease, patients with FSGS experienced similar dialysis and transplant patient survival and death-censored rate of kidney transplantation and allograft loss but higher rates of primary kidney disease recurrence.

## Conclusion

FSGS was associated with similar dialysis and transplant patient survival and death-censored first allograft loss compared to non-FSGS and other forms of glomerular disease.

## Introduction

Focal segmental glomerulosclerosis (FSGS) is one of the common causes of nephrotic syndrome [1, 2]. The prevalence of FSGS is increasing worldwide, and it is a major contributor to kidney failure [3]. Most patients are treated with immunosuppressive drugs as spontaneous remission only occurs in <10% of FSGS patients. More than 30% of patients with FSGS do not respond to treatment and progress towards kidney failure [4–6]. Such patients have a very high risk of recurrence following kidney transplantation [7–10]. However, other important clinical outcomes of FSGS patients on kidney replacement therapy (KRT), such as patient survival and kidney allograft survival, have not been well described.

This study aimed to investigate the characteristics, treatments, and outcomes of all cases of kidney failure due to FSGS in the Australian and New Zealand populations, using data from the Australia and New Zealand Dialysis and Transplant (ANZDATA) Registry.

## Materials and methods

### Patient population

All adult patients (≥18 years of age) with kidney failure who commenced KRT in Australia and New Zealand from 15[th] May 1963 to 31[st] December 2018 in the ANZDATA Registry were included. Collection and analysis of ANZDATA registry data were approved by the Nepean Blue Mountains Local Health District Human Research Ethics committee (2020/ETH00826) and the ANZDATA executive. The analyses were performed on de-identified data extracted from ANZDATA; hence written consent was not obtained.

### Exposure

The main exposure of interest was the cause of kidney failure, categorised according to whether the cause was FSGS or non-FSGS. FSGS was diagnosed by kidney biopsy. A secondary analysis categorised all patients with kidney failure due to glomerular disease (GD) into FSGS or other GD groups.

### Outcomes

Patient outcomes examined included:

1. Patient survival on dialysis, defined as the time from dialysis initiation to death. Patients were censored at the date of kidney function recovery, kidney transplantation or end of study (31 December 2018), whichever came first. Transplantation was considered a censoring event or time-varying covariate.

2. Probability of receiving a kidney transplant, defined as the time from dialysis initiation to kidney transplantation. Data were censored for death, kidney function recovery or end of study (31 December 2018), whichever came first.

3. Primary kidney disease recurrence, defined as the time from kidney transplantation to recurrence of primary kidney disease. Data were censored for death, kidney allograft loss or end of study (31 December 2018), whichever came first.

4. Allograft survival, defined as the time from first kidney transplantation to commencement of dialysis. Only survival of the first allograft was analysed for patients who received more than one transplant. Data were censored for death or end of study (31 December 2018), whichever came first.

5. Transplant patient survival, defined as the time from kidney transplantation to patient death. Data were censored for dialysis commencement or end of study (31 December 2018), whichever came first.

For outcomes 2 to 5, death was considered a competing risk. Only survival of the first allograft was analysed for patients who received more than one transplant.

## Potential confounders

The following covariates which could affect the outcomes were included in the analyses; age at initial KRT, gender, race, first KRT treatment, body mass index (BMI), smoking status, comorbidities and dialysis era or transplant era. The comorbidities recorded in ANZDATA include cerebrovascular disease, coronary artery disease, diabetes mellitus, peripheral vascular disease, and chronic lung disease.

## Statistical analysis

Summary statistics were presented as counts (percentages) for categorical data, mean ± standard deviation (SD) for normally distributed continuous data, and median [interquartile range; IQR] for non-normally distributed continuous data. Characteristics of kidney failure due to FSGS were compared to non-FSGS using Pearson's chi-square for categorical variables, two-tailed unpaired t-tests or Mann-Whitney tests for continuous variables, depending on data distribution. Survival probabilities for the six time-to-event outcomes were explored using Kaplan-Meier curves, and cumulative incidence plots were used to visualise competing risks. Multivariable Cox proportional hazards models were used to estimate the hazard ratios of FSGS vs non-FSGS for the outcomes with no competing risks. The models that included time-varying covariates, the start-stop style, were used to reflect the time-varying nature [11]. The Fine and Gray model was used to assess the association between the cause of kidney failure and cumulative incidence for outcomes with competing risks [12].

The above analysis also compared kidney failure due to FSGS versus other GD.

Statistical analysis was performed using Stata version 16.1 (StataCorp, College Station, TX). P values $<0.05$ were considered statistically significant.

## Results

### Patient characteristics

Between 15th May 1963 and 31st December 2018, 85,052 adult patients started KRT for kidney failure in Australia and New Zealand. Of these, 2,991 (3.5%) had kidney failure due to FSGS,

**Table 1. Characteristics of patients with kidney failure whose first kidney replacement therapy was dialysis in Australia and New Zealand.**

| | Total | FSGS | Non-FSGS | P value |
|---|---|---|---|---|
| | N = 82,634 | N = 2,882 | N = 79,752 | |
| **Age (years)** | 62.39 | 59.96 | 62.46 | <0.001 |
| **Gender** | | | | <0.001 |
| Male | 59% (49,083) | 67% (1,920) | 59% (47,163) | |
| **Ethnicity** | | | | <0.001 |
| White | 73% (60,181) | 78% (2,240) | 73% (57,941) | |
| Aboriginal/Torres Strait Islander | 7% (5,859) | 4% (107) | 7% (5,752) | |
| Asian | 7% (6,107) | 9% (248) | 7% (5,859) | |
| Māori | 6% (4,776) | 4% (101) | 6% (4,675) | |
| Pacific Islander | 4% (3,533) | 4% (115) | 4% (3,418) | |
| Other | 2% (1,585) | 2% (58) | 2% (1,527) | |
| Not reported | 1% (593) | 0.4% (13) | 1% (580) | |
| **KRT era** | | | | <0.001 |
| 1960–1970 | 1% (851) | 0% (8) | 1% (843) | |
| 1971–1980 | 6% (4,733) | 5% (135) | 6% (4,598) | |
| 1981–1990 | 10% (8,429) | 12% (339) | 10% (8,090) | |
| 1991–2000 | 20% (16,603) | 26% (746) | 20% (15,857) | |
| 2001–2010 | 32% (26,130) | 31% (881) | 32% (25,249) | |
| 2011–2020 | 31% (25,888) | 27% (773) | 31% (25,115) | |
| **Smoking status at KRT entry** | | | | <0.001 |
| Current | 11% (9,358) | 12% (344) | 11% (9,014) | |
| Former | 33% (27,671) | 33% (937) | 34% (26,734) | |
| Never | 41% (33,491) | 44% (1,265) | 40% (32,226) | |
| Unknown | 15% (12,114) | 12% (336) | 15% (11,778) | |
| **Diabetes mellitus** | 38% (31,129) | 14% (416) | 39% (30,713) | <0.001 |
| **Chronic lung disease** | 23% (18,638) | 18% (522) | 23% (18,116) | <0.01 |
| **Coronary artery disease** | 32% (26,851) | 22% (644) | 33% (26,207) | <0.001 |
| **Peripheral vascular disease** | 21% (16,944) | 10% (274) | 21% (16,670) | <0.001 |
| **Cerebrovascular disease** | 12% (9,747) | 8% (217) | 12% (9,530) | <0.001 |
| **BMI (kg/m$^2$)** | 28.25 | 28.7 | 28.23 | 0.008 |
| **First KRT** | | | | 0.001 |
| Haemodialysis | 70% (57,564) | 67% (1,929) | 70% (55,635) | |
| Peritoneal dialysis | 30% (25,070) | 33% (953) | 30% (24,117) | |
| **Follow-up years** | 6.66 (7.31) | 9.24 (8.36) | 6.57 (7.26) | <0.001 |

**Abbreviations**: FSGS, Focal Segmental Glomerulosclerosis; BMI, body mass index; KRT, kidney replacement therapy

including 2,882 (96.4%) who underwent dialysis and 109 (3.6%) who underwent kidney transplantation as their first KRT. The baseline characteristics of FSGS and non-FSGS are depicted in Table 1. The causes of kidney failure in patients whose first kidney replacement therapy was dialysis in Australia and New Zealand are shown in S1 Table.

S2 Table describes patient characteristics between FSGS and other GD. The baseline characteristics were comparable between the two groups. Still, diabetes mellitus was more common in the FSGS group (16% versus 10%), and there were more patients with healthy BMI in the GD group (42% versus 35%).

## Patient survival on dialysis

There were 1027/2882 deaths (35.6%) amongst FSGS patients (97 per 1000 patient-years) on dialysis compared to 40589/79752 (50.8%) amongst patients with kidney failure from non-FSGS causes (153 per 1000 patient-years). There were more cardiovascular-related deaths in the non-FSGS patients and more cancer-related deaths in the FSGS patients (p = 0.02) (S3 Table). The high cardiovascular death could be due to the higher cardiovascular risk, such as diabetes, coronary artery and peripheral vascular disease in the non-FSGS group. The median follow-up duration of patients with kidney failure due to FSGS and non-FSGS were 2.5 and 2.4 years, respectively. The median survival duration on dialysis was 6.9 years (95% CI 6.57–7.34) for FSGS patients and 4.6 years (95% CI 4.57–4.67) for non-FSGS patients. Respective unadjusted patient survival rates at 1, 5 and 10 years were 94%, 64% and 28% in the FSGS group and 88%, 46% and 17% in the non-FSGS group (Fig 1). The risk of mortality in the FSGS group was comparable to that in the non-FSGS group (adjusted hazard ratio [aHR] 0.98, 95% CI 0.90–1.06, p = 0.55). Similar results were obtained when kidney transplantation was considered as time-varying covariate (HR 0.99, 95% CI 0.92–1.07, p = 0.86 for no transplant and HR 0.97, 95% CI 0.83–1.14, p = 0.74 for transplant) (S1 Fig).

When patients with FSGS were compared to patients with kidney failure due to other GD, the mortality risk was similar (aHR 0.99, 95% CI 0.91–1.08, p = 0.87). Results remained similar when kidney transplantation was analysed as a time-varying risk modifier (HR 1.00, 95% CI 0.92–1.08, p = 0.93 for no transplant and HR 1.04, 95% CI 0.88–1.23, p = 0.63 for transplant).

## Likelihood of kidney transplantation in patients whose first KRT was dialysis

During the study period, 24,820 patients received kidney transplantation whose first KRT was dialysis, of whom 1,360/2882 (47%) were in the FSGS group, and 23,460/79,752 (29%) were in the non-FSGS group (p<0.001). The median times from dialysis commencement to kidney transplantation for the FSGS and non-FSGS groups were 1.76 years and 1.64 years,

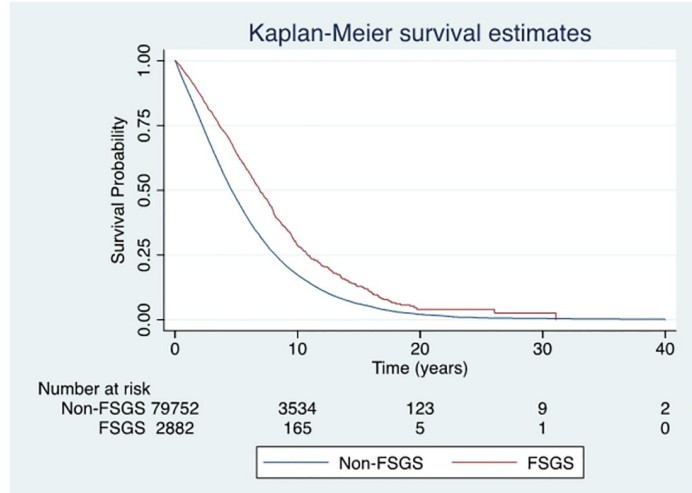

**Fig 1. Survival curves for patients with kidney failure due to FSGS or non-FSGS whose first kidney replacement therapy was dialysis in Australia and New Zealand.** The unadjusted Kaplan-Meier curve showed no significant difference between the two groups (adjusted hazard ratio 0.98, 95% CI 0.90–1.06, p = 0.55). **Abbreviations**: FSGS, Focal segmental glomerulosclerosis.

respectively. Patients with FSGS were more likely to have longer waiting times for kidney transplantation when compared to non-FSGS patients (aHR 0.92, 95% CI 0.86–0.98, p = 0.02) (S2 Fig). Considering death as a competing risk, no difference in the likelihood of kidney transplantation was observed between the two groups (SHR 0.94, 95% CI 0.87–1.01, p = 0.08). The cumulative incidence of death (S3 Fig) was appreciably higher for non-FSGS than FSGS, while the cumulative incidence of kidney transplantation was appreciably higher in the FSGS group.

Similar results were observed when comparing patients with FSGS to those with other GD (aHR 0.90, 95% CI 0.83–0.97, p = 0.004; SHR 0.93, 95% CI 0.86–1.00, p = 0.05).

### Primary kidney disease recurrence in kidney transplant recipients

During the study period, 27,238 patients received kidney transplants, of whom 1,469 were FSGS patients and 25,769 were non-FSGS patients (including those whose first KRT was dialysis and transplantation). The characteristics of patients with kidney failure due to FSGS who underwent first kidney allograft compared to those with kidney failure due to non-FSGS causes are listed in Table 2. There was more diabetes mellitus (17.9% versus 6.9%, chronic lung disease (20.2% versus 12.5%) and peripheral vascular disease (5.8% versus 2.5%) in the non-FSGS group compared to the FSGS group.

The characteristics of patients with kidney failure due to GD who received their first kidney transplantation are shown in S4 Table. Diabetes mellitus was more common in FSGS when compared to other GD (6.9% versus 4.1%).

With respect to post-transplant recurrence rates, we only compared the recurrence rates between patients with FSGS and those with kidney failure due to other GD. We did not compare FSGS versus non-FSGS as some of the non-FSGS causes of kidney failure do not recur post-transplant.

In total, 690 patients with GD had a recurrence of their primary disease, of which 163 (11.1%) were FSGS, and 527 (5.33%) were other GD. FSGS was associated with an increased risk of recurrence of primary kidney disease when compared to other forms of GD (aHR 1.67, 95% CI 1.30–2.14, p<0.001). Competing risk analysis showed similar results (SHR 1.30, 95% CI 1.05–1.60, p = 0.02).

### Kidney transplant allograft survival

The median follow-up periods for FSGS and non-FSGS patients were 6.7 years and 6.4 years, respectively. Respective unadjusted first allograft survival rates at 1, 5 and 10-years were 89%, 78% and 68% for FSGS patients and 89%, 81% and 71% for non-FSGS patients. The risk of first allograft loss was higher in patients with FSGS (aHR 1.20, 95% CI 1.04–1.37, p = 0.01) (Fig 2). However, on competing risk analysis, graft survival was similar in both groups (SHR 1.09, 95% CI 0.94–1.28, p = 0.26) (S4 and S5 Figs). Similar results were noted when recurrence of primary disease was considered as an effect modifier (with no recurrence SHR 1.09, 95% 0.93–1.27, p = 0.27 and with recurrence SHR 1.28, 95% CI 0.96–1.69, p = 0.09). The causes of allograft failure are shown in S5 Table.

Similar results were also observed when comparing FSGS patients to other GD (aHR 1.21, 95% CI 1.05–1.39, p = 0.007; SHR 1.10, 95% CI 0.92–1.31, p = 0.29).

### Kidney transplant patient survival

The unadjusted patient survival rates for transplant recipients at 1, 5 and 10 years were 98%, 94% and 83% for FSGS patients and 96%, 90% and 79% for non-FSGS patients. The risk of all-cause mortality was not significantly different between the two groups (aHR 0.92, 95%CI 0.73–1.15, p = 0.47) (Fig 3). Similarly, no difference in transplant patient survival was observed

**Table 2. Characteristics of patients undergoing first kidney transplantation for kidney failure in Australia and New Zealand.**

| | Total | FSGS | Non-FSGS | P value |
|---|---|---|---|---|
| | N = 27238 | N = 1469 | N = 25769 | |
| **Age (years)** | 43.9 | 42.3 | 44 | <0.001 |
| **Gender** | | | | <0.001 |
| Male | 60.9% (16577) | 66.4% (975) | 60.6% (15602) | |
| **Ethnicity** | | | | 0.005 |
| Caucasian | 81.7% (22260) | 81.3% (1194) | 81.8% (21066) | |
| Aboriginal/Torres Strait Islander | 2.7% (744) | 2% (29) | 2.8% (715) | |
| Asian | 8.2% (2220) | 9% (133) | 8.1% (2087) | |
| Māori | 2.7% (721) | 2.2% (32) | 2.7% (689) | |
| Pacific Islander | 2.2% (609) | 3.3% (49) | 2.2% (560) | |
| Other | 1.7% (466) | 1.8% (27) | 1.7% (439) | |
| Not reported | 0.8% (218) | 0.3% (5) | 0.8% (213) | |
| **KRT era** | | | | <0.001 |
| 1960–1970 | 2.3% (626) | 0.5% (8) | 2.4% (618) | |
| 1971–1980 | 10.6% (2877) | 5.4% (79) | 10.9% (2798) | |
| 1981–1990 | 14.9% (4060) | 15.4% (226) | 14.9% (3834) | |
| 1991–2000 | 18.2% (4952) | 20.8% (305) | 18% (4647) | |
| 2001–2010 | 25.1% (6843) | 29.6% (435) | 24.9% (6408) | |
| 2011–2020 | 28.9% (7880) | 28.3% (416) | 29% (7464) | |
| **Smoking status at KRT entry** | | | | <0.001 |
| Current | 8.2% (2237) | 8.6% (126) | 8.2% (2111) | |
| Former | 23.2% (6331) | 25.2% (370) | 23.1% (5961) | |
| Never | 45.2% (12303) | 51.5% (757) | 44.8% (11546) | |
| Unknown | 23.4% (6367) | 14.7% (1469) | 23.9% (6151) | |
| **Diabetes** | 17.2% (3986) | 6.9% (93) | 17.9% (3893) | <0.001 |
| **Chronic lung disease** | 19.8% (5395) | 12.5% (184) | 20.2% (5211) | <0.001 |
| **Coronary artery disease** | 9.7% (2182) | 8.8% (115) | 9.8% (2067) | 0.21 |
| **Peripheral vascular disease** | 5.6% (1263) | 2.5% (33) | 5.8% (1230) | <0.001 |
| **Cerebrovascular disease** | 3.2% (723) | 2% (26) | 3.3% (697) | 0.008 |
| **BMI (kg/m$^2$)** | 25.6 | 25.6 | 25.6 | 0.04 |
| **First KRT** | | | | 0.13 |
| Haemodialysis | 61.4% (16714) | 62% (911) | 61.3% (15803) | |
| Peritoneal Dialysis | 29.8% (8106) | 30.6% (449) | 29.7% (7657) | |
| Transplantation | 8.9% (2418) | 7.4% (109) | 9% (2309) | |
| **Subsequent allografts** | 13.4% (4209) | 16.3% (287) | 13.2% (3922) | 0.002 |
| *Second allograft* | 11.4% (3575) | 13.4% (236) | 11.3% (3339) | |
| *Third or more allografts* | 2.1% (634) | 2.9% (51) | 1.9% (583) | |
| **Follow-up years** Mean (95% CI) | 13.06 (12.95–13.17) | 14.12 (13.67–14.57) | 13 (12.89–13.12) | |

**Abbreviations**: FSGS, Focal Segmental Glomerulosclerosis; BMI, body mass index; KRT, kidney replacement therapy

when considering graft loss as a time-varying risk modifier (HR 0.95, 95% CI 0.76–1.19, p = 0.66 for no graft loss and HR 1.22, 95% CI 0.98–1.52, p = 0.07 for graft loss) (S6 Fig). The causes of death are shown in S6 Table.

Similar results were observed when comparing patients with FSGS to those with other forms of GD (aHR 0.96, 95%CI 0.76–1.21, p = 0.71; SHR 0.94, 95% CI 0.74–1.18, p = 0.58).

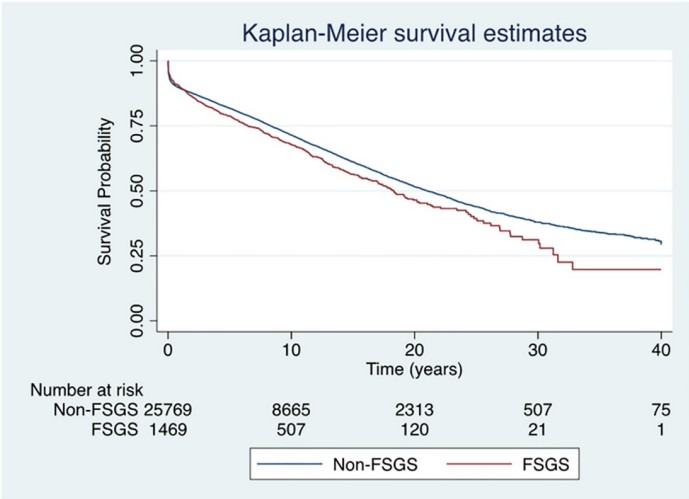

**Fig 2. First kidney allograft survival curves for patients with kidney failure due to FSGS or non-FSGS undergoing kidney transplantation in Australia and New Zealand.** The unadjusted Kaplan-Meier curve showed that first allograft loss was higher in patients with FSGS (adjusted hazard ratio was 1.20, 95% CI 1.04–1.37, p = 0.01). **Abbreviations**: FSGS, Focal segmental glomerulosclerosis.

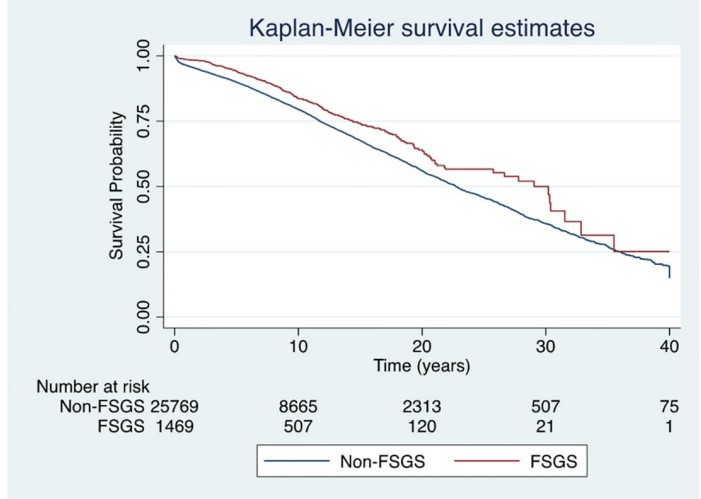

**Fig 3. Transplant patient survival curves for individuals with kidney failure due to FSGS or non-FSGS undergoing first kidney transplantation in Australia and New Zealand.** The unadjusted Kaplan-Meier curve showed no significant difference between the two groups (adjusted hazard ratio 0.92, 95% CI 0.73–1.15, p = 0.47). **Abbreviations**: FSGS, Focal segmental glomerulosclerosis.

## Discussion

This retrospective, bi-national, multicentre registry-based study showed that patients with kidney failure due to FSGS treated with dialysis have comparable probabilities of undergoing kidney transplantation, experiencing dialysis-independent kidney function recovery and surviving when compared to non-FSGS patients. Following first kidney transplantation, allograft survival and patient survival were also comparable between FSGS and non-FSGS patients.

On subgroup analysis, similar results were noted when comparing FSGS with other GD patients, except that FSGS patients had lower dialysis-independent kidney function recovery rates compared to other GD.

A previous study from the United States Renal Data System (USRDS) reported the outcomes of 84,301 kidney failure patients on KRT due to GD from 1996 to 2011 [13]. There were 34,330 FSGS patients on KRT (6.1% transplant patients). All GD subtypes were compared with IgA nephropathy. Compared to IgA nephropathy patients, the adjusted hazard ratio of mortality for FSGS patients was 1.37 (95% CI, 1.32–1.42). Similar to our study, cardiovascular disease was the most common cause of death (44.6%). The same group published (USRDS) the outcomes of GD patients post kidney transplantations [14]. There were 13,272 FSGS patients; compared to IgA nephropathy, the hazard ratio for mortality among FSGS patients was 1.57 (95% CI, 1.43–1.72). Again, cardiovascular disease was the most common cause of death. The same study reported an allograft failure rate of 32.8% in FSGS patients and an all-cause allograft failure rate of 6.09 per 100 patient-years. After excluding death as a competing risk, the hazard ratio for allograft failure in FSGS compared to IgA nephropathy was 1.20 (95% CI, 1.12–1.28). Chronic rejection was the commonest cause of death-censored cause of allograft failure. The results of our study are not directly comparable with the USRDS investigation because of differences in comparators (non-FSGS versus IgA nephropathy), differences in survival analyses (separated versus combined dialysis and kidney transplant analyses), and a high degree of missing data in the USRDS study (32%).

Approximately 2% of patients commencing dialysis for kidney failure due to FSGS experienced dialysis-independent kidney function recovery. This frequency was not different from non-FSGS or other GD patients and was similar to those previously published from ANZDATA [15, 16]. Although the likelihood of this event was not high, caution should be exercised when considering early transplantation for kidney failure patients.

In the present study, patients with kidney failure due to FSGS were more likely to receive kidney transplantation when compared to non-FSGS patients (47% versus 29%). The cumulative incidence of death was higher in the non-FSGS group, which could explain this difference. Notably, after adjusting for death as a competing risk, the rate of kidney transplantation was similar in both groups. Following kidney transplantation, patients with FSGS experienced worse graft survival than other causes of kidney failure or other forms of GD. However, this could again be explained by the high death rate observed in the non-FSGS group, given that, when death was analysed as a competing risk, the graft survival rate was similar in both groups (this was also observed when comparing FSGS with other GD). In the USRDS study, the all-cause allograft failure rate for FSGS patients was 6.09 events per 100 person-years, which was reduced to 4.41 events per 100 person-years after excluding death as a cause [14]. Death-censored allograft failure risk was higher in FSGS than in IgA nephropathy (n = 32,131, aHR 1.20, 95% CI 1.12–1.28). Using data from the UK Renal Registry, Pruthi et al. demonstrated higher allograft failure rates in patients with FSGS compared to those with ADPKD (n = 5515, aHR 2.39, 95% CI 1.78–3.22 at 10-years) [17]. Similar results were reported from the European Renal Association-European Dialysis and Transplantation Association Registry study with higher death censored allograft failure risk observed in FSGS compared with ADPKD (n = 14383, aHR 1.77 95% CI 1.50–2.08 at 15 years) [18].

One of the possible explanations for poor allograft survival with FSGS could be a higher recurrence rate of FSGS in the allograft. In the present study, the rate of recurrence of FSGS was 11% after a median follow-up period of 0.5 years. This rate is similar to what was previously reported from ANZDATA [19]. However, previously reported rates of recurrence of FSGS from other smaller studies and registries are very wide, ranging from 9% to 55% [20–25]. The most likely reason for this wide range is probably the small sample size of the reported

studies and thus insufficient power to precisely determine the incidence of recurrent FSGS. In the present study, of those who experienced a recurrence of FSGS, 71.8% progressed to graft loss after a median follow-up of 1.4 years. This is higher than reported in the literature (30%-52%) [7, 26–28].

The overall transplant patient survival in FSGS was similar to non-FSGS and other GD. Even when graft loss was considered a risk modifier, patient survival remained the same. Similar patient survival for FSGS was reported from the UK Renal Registry with 297 patients with FSGS compared to IgA nephropathy (aHR 0.95, 95% CI 0.62–1.45, p = 0.8) [17]. However, a USRDS study involving 13,272 patients reported higher mortality in patients with FSGS than those with IgA nephropathy (HR 1.57, 95% CI 1.43–1.72) [14]. This may have been due to a higher comorbidity burden observed in patients with FSGS.

The main strengths of this study were its large sample size and inclusiveness. We included all patients with FSGS receiving KRT in Australia and New Zealand during the study period, including from various centres with varying approaches to treating FSGS and kidney failure. This enhanced the external validity of our findings. However, there were some study limitations, which need to be balanced against the strengths. Limitations included heterogeneity in kidney failure care and limited depth of collected information. The ANZDATA Registry does not collect important information, such as FSGS severity, primary or secondary FSGS, proteinuria level, laboratory investigations, detailed kidney biopsy results (such as the degree of interstitial fibrosis, tubular atrophy and glomerulosclerosis), treatment protocols, side effects incurred, time from diagnosis to KRT commencement, the severity of comorbidities and socioeconomic status. Even though we adjusted for many demographic, clinical and KRT variables, residual confounding was still possible. The frequency of FSGS recurrence in kidney allografts may have been under-estimated as not all FSGS patients who experienced kidney allograft failure received a kidney biopsy. Like other registries, ANZDATA is a voluntary registry, and there is no external audit of data accuracy. As the patients' nephrologists determined the diagnosis of FSGS and comorbidities, coding bias is possible.

In conclusion, patients with FSGS had similar survival on dialysis and following kidney transplantation and similar allograft survival compared to patients with other causes of kidney failure but higher rates of primary kidney disease recurrence post-transplantation. Similar results were noted when patients with FSGS were compared with those with other forms of glomerular disease. These findings will help better inform discussions and shared decision-making between clinicians, patients with FSGS initiating KRT, and their caregivers.

## Supporting information

**S1 Table. Causes of kidney failure in patients whose first kidney replacement therapy was dialysis in Australia and New Zealand.**
(DOCX)

**S2 Table. Characteristics of patients with kidney failure due to FSGS or other glomerular disease (GD) whose first kidney replacement therapy was dialysis in Australia and New Zealand.** Abbreviations: FSGS, Focal Segmental Glomerulosclerosis; BMI, body mass index; KRT, kidney replacement therapy.
(DOCX)

**S3 Table. Causes of death in patients with kidney failure due to FSGS or non-FSGS patients whose first kidney replacement therapy was dialysis in Australia and New Zealand.** Abbreviations: FSGS, Focal Segmental Glomerulosclerosis.
(DOCX)

**S4 Table. Characteristics of patients with kidney failure due to FSGS or other Glomerular Diseases who received their first kidney allograft in Australia and New Zealand.** Abbreviations: FSGS, Focal Segmental Glomerulosclerosis; GD, Glomerular Diseases; BMI, body mass index; KRT, kidney replacement therapy.
(DOCX)

**S5 Table. Causes of kidney allograft failure in patients undergoing first kidney transplantation for kidney failure due to FSGS on Non-FSGS in Australia and New Zealand.** Abbreviations: FSGS, Focal Segmental Glomerulosclerosis. * Chronic allograft nephropathy includes chronic antibody and cell mediated rejection, calcineurin inhibitor nephrotoxicity, viral nephritis (such as BK virus), and hypertensive nephrosclerosis.
(DOCX)

**S6 Table. Causes of death in patients undergoing first kidney transplantation for kidney failure in Australia and New Zealand.** Abbreviations: FSGS, Focal Segmental Glomerulosclerosis.
(DOCX)

**S1 Fig. Survival curves for patients with kidney failure due to FSGS or Non-FSGS whose first kidney replacement therapy was dialysis in Australia and New Zealand. Transplant was considered a time varying covariate**. The survival curve between the two groups was not significantly different when kidney transplantation was considered as time-varying covariate (HR 0.99, 95% CI 0.92–1.07, p = 0.856 for no transplant and HR 0.97, 95% CI 0.83–1.14, p = 0.739 for transplant). **Abbreviations**: FSGS, Focal Segmental Glomerulosclerosis.
(TIFF)

**S2 Fig. Probability of undergoing kidney transplantation for patients with kidney failure due to FSGS or Non-FSGS treated with dialysis in Australia and New Zealand.** FSGS patients were getting transplant quicker than non-FSGS patients. The difference between the two groups was significant p < 0.001. **Abbreviations**: FSGS, Focal Segmental Glomerulosclerosis.
(TIFF)

**S3 Fig. Cause-specific cumulative incidences of undergoing kidney transplantation for patients with kidney failure due to FSGS or Non-FSGS treated with dialysis in Australia and New Zealand.** The cumulative incidence of death was higher for non-FSGS than FSGS, while the cumulative incidence of kidney transplantation was appreciably higher in the FSGS group, that is non-FSGS patients were more likely to die before receiving a transplant (SHR 1.51, 95% CI 1.43–1.59, p<0.001). **Abbreviations**: FSGS, Focal Segmental Glomerulosclerosis.
(TIFF)

**S4 Fig. First kidney allograft survival curves for patients with kidney failure due to FSGS or non-FSGS undergoing kidney transplantation in Australia and New Zealand, adjusted for death.** First kidney allograft survival was similar between both groups when death was considered as competing risk (SHR 1.09, 95% CI 0.94–1.28, p = 0.26). **Abbreviations**: FSGS, Focal segmental glomerulosclerosis.
(TIFF)

**S5 Fig. First kidney allograft survival curves for patients with kidney failure due to FSGS or non-FSGS undergoing kidney transplantation in Australia and New Zealand, adjusted for disease recurrence.** First kidney allograft survival was similar between both groups when adjusted for disease recurrence (with no recurrence SHR 1.09, 95% 0.93–1.27, p = 0.27 and

with recurrence SHR 1.28, 95% CI 0.96–1.69, p = 0.09). **Abbreviations**: FSGS, Focal segmental glomerulosclerosis.
(TIFF)

**S6 Fig. Transplant patient survival curves for individuals undergoing first kidney transplantation for kidney failure due to FSGS or non-FSGS in Australia and New Zealand, adjusted for graft loss (as time varying covariate).** There was no significant difference between the two groups when graft loss was considered as a time-varying risk modifier (HR 0.95, 95% CI 0.76–1.19, p = 0.66 for no graft loss and HR 1.22, 95% CI 0.98–1.52, p = 0.07 for graft loss). **Abbreviations**: FSGS, Focal segmental glomerulosclerosis.
(TIFF)

**S7 Fig.**
(TIFF)

**S1 Dataset.**
(ZIP)

**S1 File. Inclusivity in global research.**
(DOCX)

## Acknowledgments

We acknowledge the substantial contributions of the entire Australian and New Zealand nephrology community (physicians, surgeons, database managers, nurses, renal operators, and patients) in providing information for and maintaining the ANZDATA Registry database. The ANZDATA Registry is funded by the Australian Organ and Tissue Donation and Transplantation Authority, the New Zealand Ministry of Health, Kidney Health Australia and Better Evidence and Translation in Chronic Kidney Disease (BEAT-CKD).

## Author Contributions

**Conceptualization:** Bhadran Bose, Elasma Milanzi, David W. Johnson, Sunil V. Badve.

**Data curation:** Bhadran Bose, Elasma Milanzi.

**Formal analysis:** Bhadran Bose, Elasma Milanzi.

**Methodology:** Bhadran Bose, Elasma Milanzi, Elaine M. Pascoe, Sunil V. Badve.

**Project administration:** Bhadran Bose, Elasma Milanzi, David W. Johnson, Sunil V. Badve.

**Software:** Bhadran Bose.

**Supervision:** Elasma Milanzi, David W. Johnson, Sunil V. Badve.

**Writing – original draft:** Bhadran Bose.

**Writing – review & editing:** Bhadran Bose, Elasma Milanzi, Elaine M. Pascoe, David W. Johnson, Sunil V. Badve.

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
