## [Decision Letter · Decision Letter 0]

7 Sep 2022

PONE-D-22-23253The outcomes of patients with kidney failure due to focal segmental glomerulosclerosis (FSGS) in Australia and New Zealand: A cohort study using the Australia and New Zealand Dialysis and Transplant Registry (ANZDATA)PLOS ONE

Dear Dr. Bose,

Thank you for submitting your manuscript to PLOS ONE. After careful consideration, we feel that it has merit but does not fully meet PLOS ONE’s publication criteria as it currently stands. Therefore, we invite you to submit a revised version of the manuscript that addresses the points raised during the review process.

The manuscript has significant flaws and is not acceptable in its current form.

The study is based on data extracted from the ANZDATA Registry which was collected over a period of more than 50 years, during which, many aspects of diagnostic and clinical approach to kidney transplantation have changed, which may result in too broad heterogeneity of  information for appropriate in depth analysis.

Important data are missing for example the number of patients for each major cause of ESRD (Hypertension, Diabetes, congenital uropathies, polycystic kidney …), many of these diseases do not recur. Only 25% of the patients with “GN” had a native biopsy, the number of patients for specific type of glomerular diseases is not provided and, absent this information, analysis of the rate of recurrence between FSGS and non-FSGS or GN group is meaningless.

Although the majority of FSGS patients had a biopsy in the native disease, it is not clear how recurrence was diagnosed.

Most of the the tables  need to be streamlined into a more concise format, the current one makes it impossible for the reader to review the difference, and its significance, among the main category of data presented.

Figures need titles, and the level of significance is often lacking in the legend.

We look forward to receiving your revised manuscript.

Kind regards,

Serena Maria Bagnasco, MD

Academic Editor

PLOS ONE

Journal Requirements:

Additional Editor Comments:

The manuscript has significant flaws and is not acceptable in its current form.

The study is based on data extracted from the ANZDATA Registry which was collected over a period of more than 50 years, during which, many aspects of diagnostic and clinical approach to kidney transplantation have changed, which may result in too broad heterogeneity of information for appropriate in depth analysis.

Important data are missing for example the number of patients for each major cause of ESRD (Hypertension, Diabetes, congenital uropathies, polycystic kidney …), many of these diseases do not recur. Only 25% of the patients with “GN” had a native biopsy, the number of patients for specific type of glomerular diseases is not provided and, absent this information, analysis of the rate of recurrence between FSGS and non-FSGS or GN group is meaningless.

Although the majority of FSGS patients had a biopsy in the native disease, it is not clear how recurrence was diagnosed.

Most of the the tables need to be streamlined into a more concise format, the current one makes it impossible for the reader to review the difference, and its significance, among the main category of data presented.

Figures need titles, and the level of significance is often lacking in the legend.

Reviewers' comments:

Reviewer's Responses to Questions

**Comments to the Author**

1. Is the manuscript technically sound, and do the data support the conclusions?

Reviewer #1: No

Reviewer #2: Yes

2. Has the statistical analysis been performed appropriately and rigorously? 

Reviewer #1: No

Reviewer #2: Yes

3. Have the authors made all data underlying the findings in their manuscript fully available?

Reviewer #1: No

Reviewer #2: Yes

4. Is the manuscript presented in an intelligible fashion and written in standard English?

Reviewer #1: No

Reviewer #2: Yes

5. Review Comments to the Author

Reviewer #1: In this work, Bose et al. show the outcomes of patients with kidney failure due to focal segmental glomerulosclerosis (FSGS) in Australia and New Zealand, using a cohort from the Australia and New Zealand Dialysis and Transplant Registry (ANZDATA)

The authors used the Registry data to analyze all adult patients who received kidney replacement therapy (dialysis or kidney transplant) in Australia and New Zealand from 15th of May 1963 to 31st of December 2018. They compared the outcome between patients who had FSGS as the cause of their kidney failure and all other patients.

They identified 85,052 patients who commenced KRT during the study period. 2991 patients (3.5%) had FSGS. Compared to patients with non-FSGS, patients with FSGS

experienced similar mortality on dialysis and following kidney transplantation

The risk of first kidney allograft loss was higher in patients with FSGS. However, when death was analyzed as a competing risk, the survival in both groups was similar. Patients

with FSGS had a longer waiting time for kidney transplantation and higher risk of disease recurrence in the allograft. Compared to patients with glomerular diseases, patients with FSGS experienced similar dialysis and transplant patient survival and death-censored rate of kidney transplantation and allograft loss but higher rates of primary kidney disease recurrence.

Major concerns:

It is not clear why the authors chose to compare patients with FSGS with all other patients with any cause of ESKD. The authors did not clarify the purpose of this work.

The authors showed that only 25% of the non-FSGS group had kidney biopsy. Considering the large number of patients in this group, it is possible that some of these patients have FSGS as well.

Considering that FSGS can be a secondary disease, it is possible that the FSGS group may have had secondary FSGS, that is due another kidney injury and not primary FSGS. It is also not clear if the authors aim to study the primary FSGS only or any type of FSGS.

The authors showed most of the parameters have statistically significant P value, however, reviewing the numbers it does not seem to be the case.

Most of published data showed that the allograft survival is less in patients with FSGS due to high risk of recurrence, this study showed when death was analyzed as a competing risk, the survival in both groups was similar. It is not clear why the death is a factor in allograft survival in this group who has very high rate of recurrent disease and allograft failure due to recurrence.

The authors showed that recurrent disease is higher in the FSGS group comparing to non-FSGS group, 11% vs 2.5 %. The recurrence rate of FSGS appears much lower than any published data, which raises the concern if really all these cases were primary FSGS. Additionally, what diseases are considered to be recurred post-transplant in the non-FSGS group?

Considering that these findings are significantly different from well-established data, the authors did not show any explanation to their results.

The findings of this work are not clinically relevant and do not add much to our understanding of FSGS as compared to other causes of ESKD.

Minor concerns:

Having the tables inside the text makes the manuscript harder to read.

Reviewer #2: It is with interest that I reviewed this manuscript entitled "The outcomes of patients with kidney failure due to focal segmental glomerulosclerosis (FSGS) in Australia and New Zealand: A cohort study using the Australia and New Zealand Dialysis and Transplant Registry (ANZDATA)".

The Authors show that compared to non FSGS patients with ESRD, FSGS patients show similar survival rate on dialysis and after transplant, but higher risk of graft loss after transplant, likely related to higher rates of recurrence compared to other forms of glomerulonephritis leading to ESRD. FSGS patients are also younger and spend more time on the transplant waiting list.

The paper is well written and the results are of interest, although clarity of discussion could be improved.

The main strength and novelty of the paper is providing prognostic information on FSGS patients with ESRD: such data are not easily available in the existing literature.

I only have minor observations:

1) Table 1 (and tables in general): please use standard approach to decimals in percentages

2) Page 18: It's actually meaningless to compare recurrence rate of FSGS with non-FSGS, as the latter group includes diseases that can't recur after tx (i.e., ADPK). in addition to this analysis, the Authors correctly show that FSGS had higher recurrence rate compared to other forms of GN (supplementary materials): I would only include this analysis which is much more meaningful

3) Supplementary table 2: please provide p values: in the text (page 11) is stated that no difference are observed in the causes of death, but some of these numbers seems actually different between groups

4) Supplementary figure 3: please double check the figure: it seems that xy axis are not correct

5) Supplementary table 4: correct typo in line 2 column 4

6) Supplementary table 4: Chronic allograft nephropathy is an obsolete definition (eliminated from Banff about a decade ago). I understand that grouping different diagnosis from such different time periods can be hard, but I would suggest to explain that CAN actually includes different etiologies that have been better defined over time (chronic ab and cell mediated rejection, CNI toxicity, viral nephritis, etc)

6. PLOS authors have the option to publish the peer review history of their article (what does this mean?). If published, this will include your full peer review and any attached files.

Reviewer #1: No

Reviewer #2: No

---

## [Author Response · Author response to Decision Letter 0]

17 Oct 2022

17/10/2022

Dr Bhadran Bose

Department of Nephrology, 

Nepean Hospital

Kingswood NSW 2747, 

Australia

Email: bhadran.bose@health.nsw.gov.au

Dr Serena Maria Bagnasco

Academic Editor

PLOS ONE

Dear Dr Bagnasco

Re: Resubmission of the manuscript titled, “The outcomes of patients with kidney failure due to focal segmental glomerulosclerosis (FSGS) in Australia and New Zealand: A cohort study using the Australia and New Zealand Dialysis and Transplant Registry (ANZDATA)” – PONE-D-22-23253

We thank the reviewers and editor for their valuable feedback. We have addressed the comments raised by the editor and reviewers in this letter. In addition, we have highlighted our response to each of the comments.

Additional Editor Comments:

Comment 1: The manuscript has significant flaws and is not acceptable in its current form.

The study is based on data extracted from the ANZDATA Registry which was collected over a period of more than 50 years, during which, many aspects of diagnostic and clinical approach to kidney transplantation have changed, which may result in too broad heterogeneity of information for appropriate in-depth analysis.

Response: We agree, that many aspects of dialysis and transplantation have changed over the last 50 years, which may have confounded the findings. In order to account for this, we have adjusted for dialysis or transplant era in our analysis.

Comment 2: Important data are missing for example the number of patients for each major cause of ESRD (Hypertension, Diabetes, congenital uropathies, polycystic kidney …), many of these diseases do not recur. 

Response: We have added a table (S1 Table) on pages 1 and 2 of the supplementary material giving a break down of the causes of kidney failure in patients whose first kidney replacement therapy was dialysis in Australia and New Zealand.

Comment 3: Only 25% of the patients with “GN” had a native biopsy, the number of patients for specific type of glomerular diseases is not provided and, absent this information, analysis of the rate of recurrence between FSGS and non-FSGS or GN group is meaningless.

Response: As per S3 Table, 99.9% of FSGS patients and 78.3% of patients with other forms of glomerular disease had a native kidney biopsy.

We agree that analysis of the rates of post-transplant recurrence between FSGS and non-FSGS is fraught. Hence, we have deleted that paragraph.

 However, we feel that comparison between FSGS and other glomerular disease is useful. As such, we have added the following paragraph (page 16, line 296 to 299).

“With respect to post-transplant recurrence rates, we only compared the recurrence rates between patients with FSGS and those with kidney failure due to other GD. We did not compare FSGS versus non-FSGS as some of the non-FSGS causes of kidney failure do not recur post-transplant.”

Comment 4: Although the majority of FSGS patients had a biopsy in the native disease, it is not clear how recurrence was diagnosed.

Response: Recurrence was diagnosed based on the treating nephrologists’ reports to ANZDATA. However, the frequency of FSGS recurrence in kidney allografts may have been under-estimated as not all patients with FSGS who experienced kidney allograft failure received a kidney biopsy.

This has been stated as one of the limitations in our discussion (page 22, line 469 to 471).

“The frequency of FSGS recurrence in kidney allografts may have been under-estimated as not all patients with FSGS who experienced kidney allograft failure received a kidney biopsy.”

Comment 5: Most of the tables need to be streamlined into a more concise format, the current one makes it impossible for the reader to review the difference, and its significance, among the main category of data presented.

Response: We have streamlined the tables (Table 1, Table 2, S2 Table, S4 Table) by deleting age categories, BMI categories and number of female patients.

Comment 6: Figures need titles, and the level of significance is often lacking in the legend.

Response: Titles had already been provided for all figures (Fig 1-3, S1-S7 Fig). We have added hazard ratio with 95% confidence interval and p value to each figure.

Reviewers' comments:

Reviewer #1: 

Major concerns:

Comment 1: It is not clear why the authors chose to compare patients with FSGS with all other patients with any cause of ESKD. The authors did not clarify the purpose of this work.

Response: We chose to compare patients with FSGS with all other patients with any cause of kidney failure because this reflects a common question asked by patients (i.e. “what is the prognosis for patients with my condition compared to other patients starting dialysis or having a kidney transplant?”). Nonetheless, we acknowledge the Reviewer’s implied concern regarding the heterogeneity of all other patients with any cause of kidney failure. To address this, we have also performed a subgroup analysis comparing patients with FSGS those with other forms of glomerular disease, as this more closely approximates a “like with like” comparison. This has been mentioned in the methods section in page 5, line 105 to 108.

Comment 2: The authors showed that only 25% of the non-FSGS group had kidney biopsy. Considering the large number of patients in this group, it is possible that some of these patients have FSGS as well.

Response: We acknowledge that this is possible and have specifically stated this as one of the limitations in our discussion (page 22, line 472-473).

“As the patients’ nephrologists determined the diagnosis of FSGS and comorbidities, coding bias is possible”

Comment 3: Considering that FSGS can be a secondary disease, it is possible that the FSGS group may have had secondary FSGS, that is due another kidney injury and not primary FSGS. It is also not clear if the authors aim to study the primary FSGS only or any type of FSGS.

Response: We have already acknowledged the possibility of coding (misclassification) bias in the limitation section of the Discussion (page 22, line 471 to 473).

“Like other registries, ANZDATA is a voluntary registry, and there is no external audit of data accuracy. As the patients’ nephrologists determined the diagnosis of FSGS and comorbidities, coding bias is possible”

Comment 4: The authors showed most of the parameters have statistically significant P value, however, reviewing the numbers it does not seem to be the case.

Response: We have confirmed that all calculated p values are correct.

Comment 5: Most of published data showed that the allograft survival is less in patients with FSGS due to high risk of recurrence, this study showed when death was analyzed as a competing risk, the survival in both groups was similar. It is not clear why the death is a factor in allograft survival in this group who has very high rate of recurrent disease and allograft failure due to recurrence.

Response: Death was considered as a competing risk because death precludes observation of the event of interest, that is, if a patient dies before a graft failure, we will not be able to observe graft loss. On the other hand, we acknowledge that the differing rate of recurrence should be considered; indeed 11% of FSGS patients experienced recurrence compared to 2.5% of non- FSGS patients. We have added the results of analysis adjusting for recurrence as an effect modifier and have reported results separately for patients who experienced recurrence and those who did not (page 17, line 333 to 336).

“Similar results were noted when recurrence of primary disease was considered as an effect modifier (with no recurrence SHR 1.09, 95% 0.93-1.27, p=0.27 and with recurrence SHR 1.28, 95% CI 0.96-1.69, p=0.09).”

Comment 6: The authors showed that recurrent disease is higher in the FSGS group comparing to non-FSGS group, 11% vs 2.5 %. The recurrence rate of FSGS appears much lower than any published data, which raises the concern if really all these cases were primary FSGS. Additionally, what diseases are considered to be recurred post-transplant in the non-FSGS group?

Considering that these findings are significantly different from well-established data, the authors did not show any explanation to their results.

Response: We respectfully disagree with the reviewer. In the literature, the reported rates of recurrent FSGS in adult population are wide. In registry-based studies, the recurrence rate is between 9% to 15%(1-3) and in smaller studies it is between 17% to 55%(4-7). This is similar to what we have observed in our registry-based study. 

Comment 7: The findings of this work are not clinically relevant and do not add much to our understanding of FSGS as compared to other causes of ESKD.

Response: We respectfully disagree. It is the first time most of the outcomes are being compared between FSGS and non-FSGS in the Australian/New Zealand population. The results of this study will help better inform discussions and shared decision-making between clinicians, patients with FSGS initiating KRT, and their caregivers.

Minor concerns:

Having the tables inside the text makes the manuscript harder to read.

Response: This was done as per the journal requirements.

Reviewer #2: 

I only have minor observations:

Comment 1: Table 1 (and tables in general): please use standard approach to decimals in percentages

Response: We have made relevant changes where appropriate. Thanks

Comment 2: Page 18: It's actually meaningless to compare recurrence rate of FSGS with non-FSGS, as the latter group includes diseases that can't recur after tx (i.e., ADPK). in addition to this analysis, the Authors correctly show that FSGS had higher recurrence rate compared to other forms of GN (supplementary materials): I would only include this analysis which is much more meaningful

Response: We have deleted the paragraph comparing FSGS versus non-FSGS but included the following sentence to explain the reasoning behind the exclusion main manuscript (page 16, line 296 to 299).

“With respect to post-transplant recurrence rates, we only compared the recurrence rates between patients with FSGS and those with kidney failure due to other GD. We did not compare FSGS versus non-FSGS as some of the non-FSGS causes of kidney failure do not recur post-transplant.”

We have also deleted S5-S6 Fig.

Comment 3: Supplementary table 2: please provide p values: in the text (page 11) is stated that no difference are observed in the causes of death, but some of these numbers seems actually different between groups

Response: A p value of 0.02 was obtained from a Chi-square test and it shows there was statistically significant different between the two groups. We have changed our statement as below. We thank the reviewer for pointing this out. This has been added in page 10, line 196-197.

“There were more cardiovascular related deaths in the non-FSGS patients and more cancer related deaths in the FSGS patients (p=0.02)”

Comment 4: Supplementary figure 3: please double check the figure: it seems that xy axis are not correct

Response: We agree with the reviewer that graph reads wrongly as y axis looks inverted, hence we have deleted the Kaplan Meir plot and instead added a Nelson-Aalen cumulative hazard estimates plot which shows that the probability of undergoing transplant increases with time and then plateaus. We thank the reviewer for pointing this one out. 

Comment 5: Supplementary table 4: correct typo in line 2 column 4

Response: Typo corrected. Thanks

Comment 6: Supplementary table 4: Chronic allograft nephropathy is an obsolete definition (eliminated from Banff about a decade ago). I understand that grouping different diagnosis from such different time periods can be hard, but I would suggest to explain that CAN actually includes different etiologies that have been better defined over time (chronic ab and cell mediated rejection, CNI toxicity, viral nephritis, etc)

Response: We have added a sentence below the table explaining what chronic allograft nephropathy means in Page 9 of the supplementary material.

“Chronic allograft nephropathy includes chronic antibody and cell mediated rejection, calcineurin inhibitor nephrotoxicity, viral nephritis (such as BK virus), and hypertensive nephrosclerosis”

Along with this letter, we are resubmitting the main manuscript and the supplementary material (both clean versions and with track changes).

Once again, we thank you for reviewing our manuscript. We look forward to hearing the outcome of your editorial review in the near future.

Sincerely

Dr Bhadran Bose

On behalf of the authors

1. Allen PJ, Chadban SJ, Craig JC, Lim WH, Allen RDM, Clayton PA, et al. Recurrent glomerulonephritis after kidney transplantation: risk factors and allograft outcomes. Kidney Int. 2017;92(2):461-9.

2. Francis A, Trnka P, McTaggart SJ. Long-Term Outcome of Kidney Transplantation in Recipients with Focal Segmental Glomerulosclerosis. Clin J Am Soc Nephrol. 2016;11(11):2041-6.

3. Jiang SH, Kennard AL, Walters GD. Recurrent glomerulonephritis following renal transplantation and impact on graft survival. BMC Nephrol. 2018;19(1):344.

4. Senggutuvan P, Cameron JS, Hartley RB, Rigden S, Chantler C, Haycock G, et al. Recurrence of focal segmental glomerulosclerosis in transplanted kidneys: analysis of incidence and risk factors in 59 allografts. Pediatr Nephrol. 1990;4(1):21-8.

5. Banfi G, Colturi C, Montagnino G, Ponticelli C. The recurrence of focal segmental glomerulosclerosis in kidney transplant patients treated with cyclosporine. Transplantation. 1990;50(4):594-6.

6. Ingulli E, Tejani A. Incidence, treatment, and outcome of recurrent focal segmental glomerulosclerosis posttransplantation in 42 allografts in children--a single-center experience. Transplantation. 1991;51(2):401-5.

7. Dantal J, Baatard R, Hourmant M, Cantarovich D, Buzelin F, Soulillou JP. Recurrent nephrotic syndrome following renal transplantation in patients with focal glomerulosclerosis. A one-center study of plasma exchange effects. Transplantation. 1991;52(5):827-31.

---

## [Decision Letter · Decision Letter 1]

22 Nov 2022

PONE-D-22-23253R1The outcomes of patients with kidney failure due to focal segmental glomerulosclerosis (FSGS) in Australia and New Zealand: A cohort study using the Australia and New Zealand Dialysis and Transplant Registry (ANZDATA)PLOS ONE

Dear Dr. Bose,

Thank you for submitting your manuscript to PLOS ONE. After careful consideration, we feel that it has merit but does not fully meet PLOS ONE’s publication criteria as it currently stands. Therefore, we invite you to submit a revised version of the manuscript that addresses the points raised during the review process.

We look forward to receiving your revised manuscript.

Kind regards,

Rajendra Bhimma, PhD

Academic Editor

PLOS ONE

Additional Editor Comments :

Dear Dr Bhadran Bose

Thank you for the revisions to the manuscript on behalf of all your co-authors as well. The manuscript is greatly improved but Reviewer 2 has still some major concerns that need to be addressed.

Please address these concerns as soon as possible.

Reviewers' comments:

Reviewer's Responses to Questions

**Comments to the Author**

1. If the authors have adequately addressed your comments raised in a previous round of review and you feel that this manuscript is now acceptable for publication, you may indicate that here to bypass the “Comments to the Author” section, enter your conflict of interest statement in the “Confidential to Editor” section, and submit your "Accept" recommendation.

Reviewer #2: (No Response)

Reviewer #3: All comments have been addressed

2. Is the manuscript technically sound, and do the data support the conclusions?

Reviewer #2: Partly

Reviewer #3: Yes

3. Has the statistical analysis been performed appropriately and rigorously? 

Reviewer #2: Yes

Reviewer #3: Yes

4. Have the authors made all data underlying the findings in their manuscript fully available?

Reviewer #2: Yes

Reviewer #3: No

5. Is the manuscript presented in an intelligible fashion and written in standard English?

Reviewer #2: Yes

Reviewer #3: Yes

6. Review Comments to the Author

Reviewer #2: The Authors included several improvements to the the manuscript. However, I still have few observations

1

 Table 1: Percentage of native kidney biopsy: doesn't make much sense to compare FSGS with non-FSGS, as the latter will include a relevant proportion of diagnosis for which a renal biopsy is never performed (i.e., ADPKD); this data is meaningful only in STable 2

- Line 186: comment on significance of the data

- Line 221: is this paragraph relevant? numbers seem to be fairly low here

- Line 243: recurrence paragraph in results need major attention as no data on recurrence are actually explained here. Please amend.

- Line 322: these data were not shown anywhere in the text

Reviewer #3: All reviewer comments have been addressed and the article is suitable for publication.

I have no additional requests.

7. PLOS authors have the option to publish the peer review history of their article (what does this mean?). If published, this will include your full peer review and any attached files.

Reviewer #2: No

Reviewer #3: No

---

## [Author Response · Author response to Decision Letter 1]

23 Nov 2022

23/11/2022

Dr Bhadran Bose

Department of Nephrology, 

Nepean Hospital

Kingswood NSW 2747, 

Australia

Email: bhadran.bose@health.nsw.gov.au

Dr Rajendra Bhimma

Academic Editor

PLOS ONE

Dear Dr Bhimma

Re: Resubmission of the manuscript titled, “The outcomes of patients with kidney failure due to focal segmental glomerulosclerosis (FSGS) in Australia and New Zealand: A cohort study using the Australia and New Zealand Dialysis and Transplant Registry (ANZDATA)” – PONE-D-22-23253

We thank the reviewers and editor for their valuable feedback. We have addressed the comments raised by the reviewers in this letter. In addition, we have highlighted our response to each of the comments.

Along with this letter, we are resubmitting the main manuscript (both clean versions and with track changes).

Reviewer #2: The Authors included several improvements to the the manuscript. However, I still have few observations

1 Table 1: Percentage of native kidney biopsy: doesn't make much sense to compare FSGS with non-FSGS, as the latter will include a relevant proportion of diagnosis for which a renal biopsy is never performed (i.e., ADPKD); this data is meaningful only in STable 2

Response: We thank the reviewer for the valuable comment. We have deleted the row containing percentage of native kidney biopsy in Table 1.

- Line 186: comment on significance of the data

Response: We noted more cardiovascular deaths in non-FSGS patients, which could be since the non-FSGS group were older patients with more prevalence of diabetes and coronary artery disease.

FSGS group had more cancer-related deaths, which could be due to long term immunosuppression exposure but hard to comment as we don't have information regarding their treatment before commencing kidney replacement therapy.

- Line 221: is this paragraph relevant? numbers seem to be fairly low here

Response: We thank the reviewer for their comment, and we have deleted this paragraph

- Line 243: recurrence paragraph in results need major attention as no data on recurrence are actually explained here. Please amend.

Response: We thank the reviewer for highlighting this point. We have amended the manuscript describing the recurrence rates between FSGS and other GD. Line 266-268

“FSGS was associated with an increased risk of recurrence of primary kidney disease when compared to other forms of GD (aHR 1.67, 95% CI 1.30-2.14, p<0.001). Competing risk analysis showed similar results (SHR 1.30, 95% CI 1.05-1.60, p=0.02).”

- Line 322: these data were not shown anywhere in the text

Response: Based on the initial reviewer’s comment (from first revision) we had deleted the analysis comparing recurrence between FSGS and non-FSGS patients as it was felt to be meaningless by the reviewer. We should have deleted the sentence describing that analysis in our discussion. However, we have now changed the sentence. Line 320-322

“Following first kidney transplantation, allograft survival and patient survival were also comparable between FSGS and non-FSGS patients.”

Once again, we thank you for reviewing our manuscript. We look forward to hearing the outcome of your editorial review in the near future.

Sincerely

Dr Bhadran Bose

On behalf of the authors

---

## [Decision Letter · Decision Letter 2]

3 May 2023

PONE-D-22-23253R2The outcomes of patients with kidney failure due to focal segmental glomerulosclerosis (FSGS) in Australia and New Zealand: A cohort study using the Australia and New Zealand Dialysis and Transplant Registry (ANZDATA)PLOS ONE

Dear Dr. Bose,

Thank you for submitting your manuscript to PLOS ONE. After careful consideration, we feel that it has merit but does not fully meet PLOS ONE’s publication criteria as it currently stands. Therefore, we invite you to submit a revised version of the manuscript that addresses the points raised during the review process.

We look forward to receiving your revised manuscript.

Kind regards,

Rajendra Bhimma, PhD

Academic Editor

PLOS ONE

Additional Editor Comments:

Please see comments by reviewers.

Reviewers' comments:

Reviewer's Responses to Questions

**Comments to the Author**

1. If the authors have adequately addressed your comments raised in a previous round of review and you feel that this manuscript is now acceptable for publication, you may indicate that here to bypass the “Comments to the Author” section, enter your conflict of interest statement in the “Confidential to Editor” section, and submit your "Accept" recommendation.

Reviewer #4: All comments have been addressed

Reviewer #5: (No Response)

2. Is the manuscript technically sound, and do the data support the conclusions?

Reviewer #4: Yes

Reviewer #5: No

3. Has the statistical analysis been performed appropriately and rigorously? 

Reviewer #4: Yes

Reviewer #5: No

4. Have the authors made all data underlying the findings in their manuscript fully available?

Reviewer #4: Yes

Reviewer #5: No

5. Is the manuscript presented in an intelligible fashion and written in standard English?

Reviewer #4: Yes

Reviewer #5: Yes

6. Review Comments to the Author

Reviewer #4: All comments previously made have been satisfactorily answered.

However, the authors may wish to correct "Pacific" to "Pacific Islander" in the Ethnicity category of all relevant patients characteristics tables.

Reviewer #5: The outcomes of patients with kidney failure due to focal segmental glomerulosclerosis (FSGS) in Australia and New Zealand: A cohort study using the Australia and New Zealand Dialysis and Transplant Registry (ANZDATA)

Manuscript Number: PONE-D-22-23253R2

Comments to Authors and Editor

The authors attempt to investigate the characteristics, treatments, and outcomes of all cases of kidney failure due to FSGS in the Australian and New Zealand populations, using 2018 data from the Australia and New Zealand Dialysis and Transplant (ANZDATA) Registry.

Survivals between FSGS vs non-FSGS look very promising. However, I have concerns regarding the clarity of the presented data, as readers may struggle to replicate the described approach in other datasets. Therefore, I believe that the paper would benefit from revisions to the methodology and outcomes to enhance the understanding of the differences in survival rates between the two disease groups.

Specific comments

Number of participants: The FSGS and non-FSGS groups have a significant difference in the number of participants (with n=2882 and n=79,752, respectively). This can result in biased estimates of survival probabilities and hazard ratios, which may lead to incorrect conclusions about the relationship between the exposure and survival outcomes.

Outcomes: Outcome 1: The authors have already defined the kidney transplantation as a censor event in line 108, but not sure why transplantation was considered as censored event in line 109. I am not sure about the meaning of statement “Transplantation was considered a censoring event or time-varying covariate”, especially the time varying covariate. Outcome 2: What is the reason for investigating the second outcome, which is "dialysis-independent recovery of kidney function"? the number of cases for this condition is minimal, I believe. Outcome 3: I am also uncertain about the inclusion of kidney transplantation as a third outcome in the study. It appears to diverge from the main aim and objectives of the research, which focus on the impact of FSGS on outcomes such as death and graft failure. Outcome 4: The ANZDATA Registry records the primary renal disease at the start of the study. As such, I am uncertain how the recurrence of the primary disease was analysed using ANZDATA. With regards to Outcome 5, I am unsure of the rationale behind using the commencement of dialysis as an endpoint. Typically, the standard measure for graft survival is from the first kidney transplant until the failure of the graft. The authors noted that death was treated as a competing event, and it is unclear why death was censored for outcomes 2-5.

Statistical analysis: The statistical analysis plan used in the study lacks clarity, as the authors have mixed up standard survival models with competing risk analysis. Additionally, there is a lack of detailed information on the use of time-varying covariates in the survival analysis.

Results: The manuscript contains numerous findings, and the authors have included irrelevant outcomes, resulting in an unnecessarily lengthy document.

Timeframe: The ANZDATA 2018 data appears to be quite outdated. It is recommended that the authors use more recent ANZDATA 2022 data to obtain a more accurate depiction of the latest findings. This would enable them to present the actual outcomes of the recent data more effectively.

7. PLOS authors have the option to publish the peer review history of their article (what does this mean?). If published, this will include your full peer review and any attached files.

Reviewer #4: No

Reviewer #5: No

---

## [Author Response · Author response to Decision Letter 2]

7 Jun 2023

06/06/2023

Dr Bhadran Bose

Department of Nephrology, 

Nepean Hospital

Kingswood NSW 2747, 

Australia

Email: bhadran.bose@health.nsw.gov.au

Dr Rajendra Bhimma

Academic Editor

PLOS ONE

Dear Dr Bhimma

Re: Resubmission of the manuscript titled, “The outcomes of patients with kidney failure due to focal segmental glomerulosclerosis (FSGS) in Australia and New Zealand: A cohort study using the Australia and New Zealand Dialysis and Transplant Registry (ANZDATA)” – PONE-D-22-23253

We thank the reviewers for their comments and feedback. We had initially submitted our manuscript to PLOS ONE on August 19, 2022. Since then, we have had multiple revisions based on the comments and feedback from various reviewers. Following our latest submission, reviewer 5 has given some new comments. In this resubmission we will try to address most of the comments raised by reviewer 5. 

Reviewer #4: All comments previously made have been satisfactorily answered.

However, the authors may wish to correct "Pacific" to "Pacific Islander" in the Ethnicity category of all relevant patients characteristics tables.

RESPONSE: We thank the reviewer for this comment. We have changed the term “Pacific” to Pacific Islander” in the revised version of the manuscript.

Reviewer #5: The authors attempt to investigate the characteristics, treatments, and outcomes of all cases of kidney failure due to FSGS in the Australian and New Zealand populations, using 2018 data from the Australia and New Zealand Dialysis and Transplant (ANZDATA) Registry.

Survivals between FSGS vs non-FSGS look very promising. However, I have concerns regarding the clarity of the presented data, as readers may struggle to replicate the described approach in other datasets. Therefore, I believe that the paper would benefit from revisions to the methodology and outcomes to enhance the understanding of the differences in survival rates between the two disease groups.

RESPONSE: We thank the reviewer for this comment. We are unsure which aspect of the methodology and outcomes the reviewer would like us to revise. This manuscript's methodology and outcomes are similar to the previously published ANZDATA study.

Yang WL, Bose B, Zhang L, McStea M, Cho Y, Fahim M, et al. Long-term outcomes of patients with end-stage kidney disease due to membranous nephropathy: A cohort study using the Australia and New Zealand Dialysis and Transplant Registry. PLoS One. 2019;14(8):e0221531

Zhang L, Liu X, Pascoe EM, Badve SV, Boudville NC, Clayton PA, et al. Long-term outcomes of end-stage kidney disease for patients with IgA nephropathy: A multi-centre registry study. Nephrology (Carlton). 2016;21(5):387-96.

Tang W, Bose B, McDonald SP, Hawley CM, Badve SV, Boudville N, Brown FG, Clayton PA, Campbell SB, Peh CA, Johnson DW. “The outcomes of patients with end-stage renal disease and ANCA-associated vasculitis in Australia and New Zealand.” Clinical Journal of the American Society of Nephrology. Clin J Am Soc Nephrol. 2013 May;8(5):773-80. doi: 10.2215/CJN.08770812. Epub 2013 Jan 24 

Specific comments

Number of participants: The FSGS and non-FSGS groups have a significant difference in the number of participants (with n=2882 and n=79,752, respectively). This can result in biased estimates of survival probabilities and hazard ratios, which may lead to incorrect conclusions about the relationship between the exposure and survival outcomes.

RESPONSE: We thank the reviewer for this comment. It is not surprising that the ANZDATA registry has significantly more patients with kidney failure secondary to non-FSGS than FSGS as FSGS is rare condition. To avoid any bias, the following covariates which could affect the outcomes were included in the analyses; age at initial KRT, gender, race, first KRT treatment, body mass index, smoking status, comorbidities and dialysis era or transplant era. This is described in page 6, line 134 to 136.

Outcomes: Outcome 1: The authors have already defined the kidney transplantation as a censor event in line 108, but not sure why transplantation was considered as censored event in line 109. I am not sure about the meaning of statement “Transplantation was considered a censoring event or time-varying covariate”, especially the time varying covariate. 

RESPONSE: Thanks for the comments. Our full dataset includes all patients with kidney failure needing renal replacement therapy. For analysing the patient survival on dialysis (outcome 1), we censored all patients in whom kidney transplantation was their first renal replacement therapy and only included patients whose first RRT was dialysis. In patients on dialysis as their first renal replacement therapy, subsequent transplantation was considered a censoring event or time-varying covariate. This is described in page 5 line 107 to 110.

Outcome 2: What is the reason for investigating the second outcome, which is "dialysis-independent recovery of kidney function"? the number of cases for this condition is minimal, I believe. 

RESPONSE: Thanks for the comments. We have deleted this outcome from out manuscript.

Outcome 3: I am also uncertain about the inclusion of kidney transplantation as a third outcome in the study. It appears to diverge from the main aim and objectives of the research, which focus on the impact of FSGS on outcomes such as death and graft failure. 

RESPONSE: Thanks for the comments. We respectfully disagree with the reviewer. The likelihood of transplantation is a very important and clinically relevant outcome. We strongly feel that it should be reported.

Outcome 4: The ANZDATA Registry records the primary renal disease at the start of the study. As such, I am uncertain how the recurrence of the primary disease was analysed using ANZDATA. 

RESPONSE: Recurrence of primary disease is entered in the ANZDATA registry by the individual transplant unit or the treating nephrologist.

With regards to Outcome 5, I am unsure of the rationale behind using the commencement of dialysis as an endpoint. Typically, the standard measure for graft survival is from the first kidney transplant until the failure of the graft. The authors noted that death was treated as a competing event, and it is unclear why death was censored for outcomes 2-5.

RESPONSE: We thank the reviewer for the comment. The rationale for using the commencement of dialysis as an endpoint is that the patient will only be started on dialysis if there is graft failure. Otherwise, there wouldn’t be any other reason for starting dialysis. 

Statistical analysis: The statistical analysis plan used in the study lacks clarity, as the authors have mixed up standard survival models with competing risk analysis. Additionally, there is a lack of detailed information on the use of time-varying covariates in the survival analysis.

RESPONSE: We respectfully disagree with the reviewer. The statistical section of this manuscript was written, and the analysis was done by a qualified biostatistician (EM) along with a senior statistician (EP). We strongly feel that the statistical aspect of this manuscript is well described.

Results: The manuscript contains numerous findings, and the authors have included irrelevant outcomes, resulting in an unnecessarily lengthy document.

RESPONSE: We respectfully disagree with the reviewer. The findings of this study and the outcomes described are clinically relevant for clinicians and patients. The findings of this study give a better understanding of the outcomes of this patient group. Registry studies are key in providing long-term data in managing rare diseases such as FSGS, and we feel that our study, similar to other ANZDATA studies, describes long-term outcomes of patients with kidney failure.

Timeframe: The ANZDATA 2018 data appears to be quite outdated. It is recommended that the authors use more recent ANZDATA 2022 data to obtain a more accurate depiction of the latest findings. This would enable them to present the actual outcomes of the recent data more effectively.

RESPONSE: In this study, we have included patients who started RRT from 1963 to Dec 2018. However, the extracted data included follow-up until July 2021. We strongly feel that updating the data to 2022 will not make any difference in the findings of the study, especially as there hasn’t been any significant change in practice as far as management of patients with kidney failure or FSGS is concerned.

Once again, we thank you for reviewing our manuscript.

---

## [Decision Letter · Decision Letter 3]

19 Oct 2023

The outcomes of patients with kidney failure due to focal segmental glomerulosclerosis (FSGS) in Australia and New Zealand: A cohort study using the Australia and New Zealand Dialysis and Transplant Registry (ANZDATA)

PONE-D-22-23253R3

Dear Dr. Bose,

We’re pleased to inform you that your manuscript has been judged scientifically suitable for publication and will be formally accepted for publication once it meets all outstanding technical requirements.

Kind regards,

Rajendra Bhimma, PhD

Academic Editor

PLOS ONE

Additional Editor Comments (optional):

Thank you for your submission. The article have been reviewed by three reviewers and their concerns have been addressed. Whilst one of the reviewers has provided additional comments alongside their decision, these are not required revisions for your manuscript.

Reviewers' comments:

Reviewer's Responses to Questions

**Comments to the Author**

1. If the authors have adequately addressed your comments raised in a previous round of review and you feel that this manuscript is now acceptable for publication, you may indicate that here to bypass the “Comments to the Author” section, enter your conflict of interest statement in the “Confidential to Editor” section, and submit your "Accept" recommendation.

Reviewer #5: (No Response)

Reviewer #6: All comments have been addressed

2. Is the manuscript technically sound, and do the data support the conclusions?

Reviewer #5: No

Reviewer #6: Yes

3. Has the statistical analysis been performed appropriately and rigorously? 

Reviewer #5: No

Reviewer #6: Yes

4. Have the authors made all data underlying the findings in their manuscript fully available?

Reviewer #5: No

Reviewer #6: Yes

5. Is the manuscript presented in an intelligible fashion and written in standard English?

Reviewer #5: No

Reviewer #6: Yes

6. Review Comments to the Author

Reviewer #5: The outcomes of patients with kidney failure due to focal segmental glomerulosclerosis (FSGS) in Australia and New Zealand: A cohort study using the Australia and New Zealand Dialysis and Transplant Registry (ANZDATA)

Manuscript Number: PONE-D-22-23253R2

Comments to Authors and Editor

Authors have revised the article as suggested and answered my questions and comments partially. I believe, there are some scopes to improve the study prior publishing.

Number of participants FSGS (n=2882) vs non-FSGS (n=79,752)

RESPONSE: We thank the reviewer for this comment. It is not surprising that the

ANZDATA registry has significantly more patients with kidney failure secondary to

non-FSGS than FSGS as FSGS is rare condition. To avoid any bias, the following

covariates which could affect the outcomes were included in the analyses; age at initial

KRT, gender, race, first KRT treatment, body mass index, smoking status,

comorbidities and dialysis era or transplant era. This is described in page 6, line 134 to

136.

Certainly, I acknowledge that there is a scope to reduce the confounding bias by adjust the variables in the multivariate models, as you have mentioned in your responses. However, it is worthwhile to consider employing the matching process, specifically coarsened exact matching, as it can enhance the accuracy and precision of the results by mitigating the confounding bias. This methodology has the capacity to provide highly efficient and precise results that can significantly benefit the community.

Outcome 3

RESPONSE: Recurrence of primary disease is entered in the ANZDATA registry by

the individual transplant unit or the treating nephrologist.

I would like to verify if there is any documentation regarding the recurrence of primary disease within the ANZDATA database. It is certain that all hospitals in Australia and New Zealand have willingly contributed to ensure the data quality in ANZDATA. However, upon my search, I was unable to locate any information regarding the recurrence of primary disease in the database. It is possible that this parameter has been introduced recently or is not currently included in the ANZDATA data dictionary.

Statistical analysis plan:

RESPONSE: We respectfully disagree with the reviewer. The statistical section of this

manuscript was written, and the analysis was done by a qualified biostatistician (EM)

along with a senior statistician (EP). We strongly feel that the statistical aspect of this

manuscript is well described.

I have great confidence that qualified biostatisticians have been involved in this study, and I trust that they possess extensive knowledge regarding survival analysis and competing risk analysis. However, I would like to express that the provided information was insufficient for constructing statistical models. For instance:

• The selection of variables in the multivariate models is a critical aspect that should be justified throughout the model-building process.

• Each model should adhere to certain assumptions that need to be addressed.

• Assessing the goodness of fit of the model is essential to determine the validity of the findings. Additionally, considering the C-statistic for each multivariate model would provide insights into the model's predictive accuracy.

Taking these factors into consideration would significantly strengthen the study, in my opinion.

Results

RESPONSE: We respectfully disagree with the reviewer. The findings of this study and

the outcomes described are clinically relevant for clinicians and patients. The findings

of this study give a better understanding of the outcomes of this patient group. Registry

studies are key in providing long-term data in managing rare diseases such as FSGS,

and we feel that our study, similar to other ANZDATA studies, describes long-term

outcomes of patients with kidney failure.

I could be mistaken, but it might be worthwhile to investigate the reporting of both adjusted and unadjusted hazard ratios. For instance, the authors presented unadjusted hazard ratios derived from Kaplan-Meier curves, while mentioning the adjusted hazard ratios in parentheses at various points.

Lines 203-204: The unadjusted Kaplan-Meier curve showed no significant difference between the two groups (adjusted hazard ratio 0.98, 95% CI 0.90-1.06, p=0.55).

Lines 289-290: The unadjusted Kaplan-Meier curve showed that first allograft loss was higher in patients 290 with FSGS (adjusted hazard ratio was 1.20, 95% CI 1.04-1.37, p=0.01)

Lines 311-312 The unadjusted Kaplan-Meier curve showed no significant difference between the two groups (adjusted hazard ratio 0.92, 95% CI 0.73-1.15, p=0.47)

…………

Furthermore, a majority of the findings appear to be reported based on Kaplan-Meier curves, although it is unclear whether these curves were adjusted or unadjusted. It would be valuable to determine if any adjusted hazard ratios were reported from Cox proportional hazards models. If such results are indeed available, it would be helpful to refer to the model table references provided in the text.

Reviewer #6: (No Response)

7. PLOS authors have the option to publish the peer review history of their article (what does this mean?). If published, this will include your full peer review and any attached files.

Reviewer #5: No

Reviewer #6: No

---

## [Editor Report · Acceptance letter]

24 Oct 2023

PONE-D-22-23253R3 

The outcomes of patients with kidney failure due to focal segmental glomerulosclerosis (FSGS) in Australia and New Zealand: A cohort study using the Australia and New Zealand Dialysis and Transplant Registry (ANZDATA) 

Dear Dr. Bose:

I'm pleased to inform you that your manuscript has been deemed suitable for publication in PLOS ONE. Congratulations! Your manuscript is now with our production department. 

Kind regards, 

on behalf of

Professor Rajendra Bhimma 

Academic Editor

PLOS ONE